# ANALYZING THE EFFECTS OF CLASSIFIER LIPSCHITZNESS ON EXPLAINERS

## ABSTRACT

Machine learning methods are getting increasingly better at making predictions, but at the same time they are also becoming more complicated and less transparent. As a result, explanation methods are often relied on to provide interpretability to these complicated and often *black-box* prediction models. As crucial diagnostics tools, it is important that these explainer methods themselves are reliable. In this paper we focus on one particular aspect of reliability, namely that an explainer should give similar explanations for similar data inputs. We formalize this notion by introducing and defining *explainer astuteness*, analogous to astuteness of classifiers. Our formalism is inspired by the concept of *probabilistic Lipschitzness*, which captures the probability of local smoothness of a function. For a variety of explainers (e.g., SHAP, RISE, CXPlain, PredDiff), we provide lower bound guarantees on the astuteness of these explainers given the Lipschitzness of the prediction function. These theoretical results imply that locally smooth prediction functions lend themselves to locally robust explanations. We evaluate these results empirically on simulated as well as real datasets.

## 1 INTRODUCTION

Machine learning models have improved over time at prediction and classification, especially with the advances made in deep learning and availability of large amounts of data. These gains in predictive power have often been achieved using increasingly complex and *black-box* models. This has led to significant interest in, and a proliferation of, *explanation models* that provide explanations for the predictions made by these black-box models. We focus our work in analyzing the behaviour of these explanation models. In particular, we explore when are explainers reliable by analyzing the connection between the robustness of explainer methods and the smoothness of the black-box functions they are trying to explain. We propose and formally define *explainer astuteness* – a property of explainers which captures the probability that a given method provides similar explanations to similar data points. We then provide a theoretical way to connect this explainer astuteness to the *probabilistic Lipschitzness* of the black-box function that is being explained. Since probabilistic Lipschitzness is a measure of the probability that a function is smooth in a local neighborhood, our results demonstrate how the smoothness of the black-box function itself impacts the astuteness of the explainer. This implies that *enforcing smoothness on black-box functions lends them to more robust explanations.*

**Related Work.** A wide variety of explainers have been proposed in the literature (Guidotti et al., 2018; Arrieta et al., 2020). Explainers can broadly be categorized as feature attribution or feature selection explainers. Feature attribution explainers provide continuous valued importance scores to each of the input features, while feature selection explainers provide binary decisions on whether a feature is important or not. Some popular feature attribution explainers can be viewed through the lens of shapley values such as SHAP (Lundberg & Lee, 2017), LIME (Ribeiro et al., 2016) and LIFT (Shrikumar et al., 2016). Some models such as CXPlain (Schwab & Karlen, 2019), PredDiff (Zintgraf et al., 2017) and feature ablation explainers (Lei et al., 2018) calculate feature attributions by simulating individual feature removal, while other methods such as RISE (Petsiuk et al., 2018) calculate the mean effect of a feature's presence to attribute importance to it. Feature selection methods, on the other hand include individual selector approaches such as L2X (Chen et al., 2018) and INVASE (Yoon et al., 2018), and group-wise selection approaches such as gI (Masoomi et al., 2020). These models, while seemingly diverse have been shown to have striking underlying simi-

larities, for example Lundberg & Lee (2017) unify six different explanation models under a single umbrella. Recently, Covert et al. (2020) went a step ahead and combined 25 existing methods under the overall class of *removal-based explanation models*.

Similarly, there has been a recent increase in research focused on analyzing the behaviour of these explanation models themselves in ways similar to how classification models have been analyzed. Recent work has focused on dissecting various properties of explainers. Yin et al. (2021) propose stability and sensitivity (Yin et al., 2021) as measures of faithfulness of explainers to the decision-making process of the black-box model. Li et al. (2020) explore connections between local explainability and model generalization. Ghorbani et al. (2019) poke at the robustness of explainers to systemic and adversarial perturbations. Alvarez-Melis & Jaakkola (2018) empirically show that robustness, in the sense that explainers should provide similar explanations for similar inputs, is a desirable property and how forcing this property yields better explanations. Recently, Agarwal et al. (2021) explore the robustness of LIME (Ribeiro et al., 2016) and SmoothGrad (Smilkov et al., 2017), and prove that for these two methods their robustness is related to the maximum value of the gradient of the predictor function. Our work is closely related to Alvarez-Melis & Jaakkola (2018) and Agarwal et al. (2021) on explainer robustness. However, as opposed to only forcing explainers to be robust themselves (Alvarez-Melis & Jaakkola, 2018), our theoretical results suggest that ensuring robustness of explanations also depends on the smoothness of the black-box function that is being explained. Our results are complementary to the results obtained by Agarwal et al. (2021) in that our theorems cover a wider variety of explainers (see contributions below). We further relate robustness to probabilistic Lipschitzness of blackbox functions which is a quantity that can be empirically estimated.

**Contribution:**

- We formalize and define *explainer astuteness* which captures the probability that a given explainer provides similar explanations to similar points. This formalism allows us to theoretically analyze robustness properties of explainers.
- We provide theoretical results that connect astuteness of three classes of explainers; shapley value based (e.g. SHAP), explainers that simulate mean effect of features (e.g. RISE), and explainers that simulate individual feature removal (e.g. CXPlain), to the smoothness of the black-box function. Our results suggest that enforcing Lipschitzness on black-box functions can result in explainers providing more astute explanations. Formally our theorems establish a lower bound on explainer astuteness that depends on the Lipschitzness of the black-box function and square root of data dimensionality. Figure 1 summarizes this main contribution of our work.
- We demonstrate experimentally that this lower bound indeed holds in practice by comparing the astuteness predicted by our theorems to the observed astuteness on simulated and real datasets.

The remainder of this paper is organized as follows: we first provide background on the type of explanation methods we restrict ourselves to, i.e. *removal based feature explainers*, and the summary of notation used in this paper in Section 2. We then provide formal definitions and theorems with proofs in Section 3. Finally, we conclude the paper by providing experimental results in Section 4.

## 2 BACKGROUND AND NOTATIONS

### 2.1 REMOVAL-BASED FEATURE EXPLAINERS

As mentioned in Section 1 a wide variety of explainer methods has been introduced. Owing to this diversity, in this work, we concern ourselves with *removal-based feature attribution explainers* as defined by Covert et al. (2020). Removal based feature attribution explainers are methods that define a feature's influence through the impact of removing it from a model and assign continuous valued scores to each feature signifying its importance. This includes popular approaches such as SHAP and SHAP variants including KernelSHAP, LIME, DeepLIFT (Lundberg & Lee, 2017), mean effect based methods such as RISE (Petsiuk et al., 2018), and individual effects based methods such as CXPlain (Schwab & Karlen, 2019), PredDiff (Zintgraf et al., 2017), permutation tests (Strobl et al., 2008), and feature ablation explainers (Lei et al., 2018). All of these methods simulate feature removal either explicitly or implicitly. For example, SHAP explicitly considers effect of using subsets

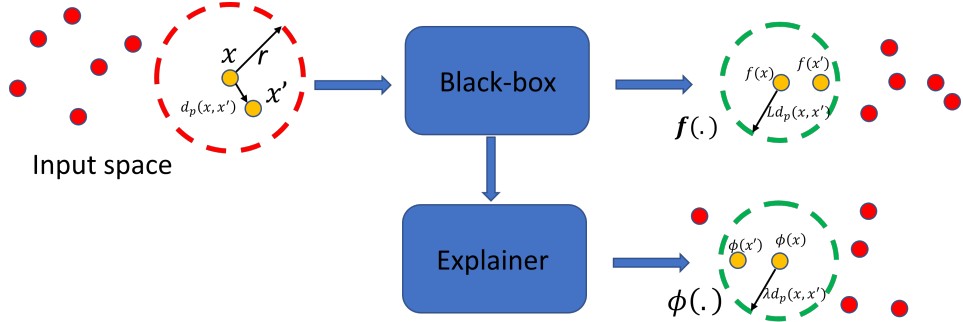

Figure 1: In this figure we visualize the implication of our theoretical results. For a prediction function that is locally *probabilistically Lipschitz* with a constant $L$, the predictions for any two points $x, x'$ such that $d_p(x, x') \leq r$ are within $Ld_p(x, x')$ distance from each other with some probability $1 - \alpha$. Given such a prediction function, the explanation for the same data points are also expected to be within $\lambda d_p(x, x')$ of each other where $\lambda \geq CL\sqrt{d}$ where C is a constant, with the same probability $1 - \alpha$.

that include a feature as compared to the effect of removing that feature from the subset. RISE removes subsets of features while always keeping the feature that is being evaluated, and estimates the average effect of keeping that feature when other features are randomly removed. CXPlain explicitly considers the impact of removing a feature on the loss function used in training the predictor function.

## 2.2 NOTATION

Before formalizing explainer astuteness and our theoretical results, we explain the notations used in the rest of the paper. We denote d-dimensional input data as $x \in \mathbb{R}^d$, from a given dataset $\mathcal{D}$. The black-box predictor function is denoted by $f$, where $f(x)$ is the prediction given $x$, this function is assumed to have been trained on the given $\mathcal{D}$. The explainer is represented by a function $\phi$ where $\phi(x) \in \mathbb{R}^d$ is the feature attribution vector representing attributions for all features in $x$ while $\phi_i(x) \in \mathbb{R}$ is the attribution for the $i^{th}$ feature. To simulate the presence or absence of features in a given subset of features, we use an indicator vector $z \in \{0, 1\}^d$, where $z_i = 1$ when the $i^{th}$ feature is present in the subset. To indicate we are only using subsets where feature $z_i = 1$, we use $z_{+i}$; and to indicate only using subsets where feature $z_i = 0$, we use $z_{-i}$. Lastly, the $p$-norm induced distance between any two points $x, x'$ is denoted by $d_p(x, x')$.

## 3 EXPLAINER ASTUTENESS

Our main interest is in defining a metric that can capture the difference in explanations provided by an explainer to points that are close to each other in the input space. The same question has been asked for classifiers and Bhattacharjee & Chaudhuri (2020) came up with the concept of *Astuteness* of classifiers, which captures the probability that similar points are assigned the same label by a classifier. Formally they provide the following definition:

**Definition 1.** *Astuteness of Classifiers (Bhattacharjee & Chaudhuri, 2020): The astuteness of a classifier $f$ over dataset $\mathcal{D}$, denoted as $A_r(f, \mathcal{D})$ is the probability that $\forall x, x' \in \mathcal{D}$ such that $d(x, x') \leq r$ the classifier will predict the same label.*

$$A_r(f, \mathcal{D}) = \mathbb{P}_{x, x' \sim \mathcal{D}}[f(x) = f(x')|d(x, x') \leq r] \quad (1)$$

The obvious difference in trying to adapt this definition of astuteness to explainers is that explanations for nearby points do not have to be *exactly* the same. Keeping this in mind, we propose and formalize *explainer astuteness*, as the probability that the explainer method assigns *similar* explanations to similar points. The formal definition is as follows:

**Definition 2.** *Explainer astuteness: The* explainer astuteness *of an explainer $E$ over dataset $\mathcal{D}$, denoted as $A_{r,\lambda}(E, \mathcal{D})$ is the probability that $\forall x, x' \in \mathcal{D}$ such that $d_p(x, x') \leq r$ the explainer $E$ will provide explanations that are at most $\lambda * d_p(x, x')$ away from each other, where $\lambda \geq 0$*

$$A_{r,\lambda}(E, \mathcal{D}) = \mathbb{P}_{x,x' \sim \mathcal{D}}[d_p(\phi(x), \phi(x)) \leq \lambda * d_p(x, x') \mid d_p(x, x') \leq r] \tag{2}$$

A critical observation about definition 2 is that it not only relates to the previously defined notion of classifier astuteness, but also connects to the concept of *probabilistic Lipschitzness*. Probabilistic Lipschitzness captures the probability of a function being locally smooth given a radius $r$. It is specially useful for capturing a notion of smoothness of complicated neural network functions for which enforcing global and deterministic Lipschitzness is difficult. Mangal et al. (2020) formally defined probabilistic Lipchitzness as follows:

**Definition 3.** *Probabilistic Lipschitz (Mangal et al., 2020): Given $0 \leq \alpha \leq 1$, $r \geq 0$, a function $f : \mathbb{X} \to \mathbb{R}$ is probabilistically Lipschitz with a constant $L \geq 0$ if*

$$\mathbb{P}_{x,x' \sim D}(d_p(f(x), f(x')) \leq L * d_p(x, x') \mid d_p(x, x') \leq r) \geq 1 - \alpha \tag{3}$$

Notice the similarity between definitions 2 and 3. This very correspondence forms the basis of our motivation to explore the connection between the Lipschitzness of the black-box prediction function, and the astuteness of the explainer, which we explore theoretically in this section and experimentally in 4.

### 3.1 Theoretical bounds of Astuteness

A cursory comparison between equation 2 and equation 3 hints at the two concepts being related to each other. In fact, explainer astuteness can be viewed as probabilistic Lipschitzness of the explainer when it is viewed as a function with a Lipschitz constant $\lambda$. However, a much more interesting question to explore is how the astuteness of explainers is connected to the Lipschitzness of the blackbox model they are trying to explain. We introduce and prove the following theorems which provide theoretical bounds that connect the Lipschitz constant $L$ of the blackbox model to the astuteness of various explanation models including SHAP (Lundberg & Lee, 2017), RISE (Petsiuk et al., 2018), and methods that simulate individual feature removal such as CXPlain (Schwab & Karlen, 2019).

#### 3.1.1 Astuteness of SHAP

SHAP (Lundberg & Lee, 2017) is one of the most popular feature attribution based explanation models in use today. Lundberg & Lee (2017) unify 6 existing explanation approaches within the SHAP framework. Each of these explanation approaches (including LIME, DeepLIFT, and kernelSHAP) can be viewed as approximations of SHAP, since SHAP in its theoretical form is difficult to calculate. However, in this section we use the theoretical definition of SHAP to establish bounds on astuteness.

For a given data point $x \in \mathcal{X}$ and a prediction function $f$ the feature attribution provided by SHAP for the $i^{th}$ feature is given by:

$$\phi_i(x) = \sum_{z_{-i}} \frac{|z_{-i}|!(d - |z_{-i}| - 1)!}{d!}[f(x \odot z_{+i}) - f(x \odot z_{-i})] \tag{4}$$

**Theorem 1.** *(Astuteness of SHAP) For a given $r \geq 0$ and $0 \leq \alpha \leq 1$, and a trained predictive function $f$ that is probilistic Lipschitz with a constant $L$ with probability at least $1 - \alpha$ at radius $r$. Then for SHAP explainers we have astuteness $A_{r,\lambda} \geq 1 - \alpha$ for $\lambda \geq 2\sqrt[p]{d}L$.*

*Proof.* Given input $x$ and another input $x'$ s.t. $d(x, x') \leq r$. And letting $\frac{|z_{-i}|!(d-|z_{-i}|-1)!}{d!} = C_z$. Using equation 4 we can write,

$d_p(\phi_i(x), \phi_i(x')) = d_p(\sum_{z_{-i}} C_z[f(x \odot z_{+i}) - f(x \odot z_{-i})], \sum_{z_{-i}} C_z[f(x' \odot z_{+i}) - f(x' \odot z_{-i})])$

Using the fact that $d_p(x,y) = ||x-y||_p$ where $||.||_p$ is the $p$-norm, the right hand side gives us,

$$d_p(\phi_i(x), \phi_i(x')) = ||\sum_{z_{-i}} C_z[f(x \odot z_{+i}) - f(x \odot z_{-i})] - \sum_{z_{-i}} C_z[f(x' \odot z_{+i}) - f(x' \odot z_{-i})]||_p$$

Combining the two sums and re-arranging,

$$
\begin{aligned}
d_p(\phi_i(x), \phi_i(x')) &= ||\sum_{z_{-i}} C_z[f(x \odot z_{+i}) - f(x \odot z_{-i}) - f(x' \odot z_{+i}) + f(x' \odot z_{-i})]||_p \\
&= ||\sum_{z_{-i}} C_z[f(x \odot z_{+i}) - f(x' \odot z_{+i}) + f(x' \odot z_{-i}) - f(x \odot z_{-i})]||_p
\end{aligned}
\tag{5}
$$

Using triangular inequality on the R.H.S twice,

$$
\begin{aligned}
d_p(\phi_i(x), \phi_i(x')) &\leq ||\sum_{z_{-i}} C_z[f(x \odot z_{+i}) - f(x' \odot z_{+i})]||_p + ||\sum_{z_{-i}} C_z[f(x' \odot z_{-i}) - f(x \odot z_{-i})]||_p \\
&\leq \sum_{z_{-i}} C_z||f(x \odot z_{+i}) - f(x' \odot z_{+i})||_p + \sum_{z_{-i}} C_z||f(x' \odot z_{-i}) - f(x \odot z_{-i})||_p
\end{aligned}
\tag{6}
$$

We can replace each value inside the sums in equation 6 with the maximum value across either sums. Doing so would still preserve the inequality in equation 6, as the sum of $n$ values is always less than the maximum among those summed $n$ times. Without loss of generality let us assume this maximum is $|f(x \odot z^*_{+i}) - f(x' \odot z^*_{+i})|$ for some particular $z^*$. This gives us:

$$d_p(\phi_i(x), \phi_i(x')) \leq ||f(x \odot z^*_{+i}) - f(x' \odot z^*_{+i})||_p \sum_{z_{-i}} C_z + ||f(x \odot z^*_{+i}) - f(x' \odot z^*_{+i})||_p \sum_{z_{-i}} C_z \tag{7}$$

However, $\sum_{z_{-i}} C_z = \sum_{z_{-i}} \frac{|z_{-i}|!(d-|z_{-i}|-1)!}{d!} = 1$, which gives us,

$$d_p(\phi_i(x), \phi_i(x')) \leq 2||f(x \odot z^*_{+i}) - f(x' \odot z^*_{+i})||_p = 2d_p(f(x \odot z^*_{+i}), f(x' \odot z^*_{+i})) \tag{8}$$

Using the fact that $f$ is probabilistic Lipschitz with given some constant $L \geq 0$, $d_p(x,x') \leq r$ and $d_p(x \odot z^*_{+i}, x' \odot z^*_{+i}) \leq d_p(x,x')$. From the definition of probablistic Lipschitz we get:

$$P[d_p(f(x \odot z^*_{+i}), f(x' \odot z^*_{+i})) \leq L * d_p(x,x')] \geq 1 - \alpha$$
$$\Rightarrow P[2d_p(f(x \odot z^*_{+i}), f(x' \odot z^*_{+i})) \leq 2L * d_p(x,x')] \geq 1 - \alpha$$

Since equation 8 establishes that $d_p(\phi_i(x), \phi_i(x')) \leq 2d_p(f(x \odot z^*_{+i}), f(x' \odot z^*_{+i}))$, the below inequality can be now established:

$$P[d_p(\phi_i(x), \phi_i(x')) \leq 2L * d_p(x,x')] \geq 1 - \alpha \tag{9}$$

Note that equation 9 is true for each feature $i \in \{1, ..., d\}$. To conclude our proof, we note that $d_p(\phi(x), \phi(x')) \leq \sqrt[p]{d} * \max_i d_p(\phi_i(x), \phi_i(x'))$ [1]. Utilizing this in equation 9, assuming the $i$ is the one corresponding to the maximum difference, gives us:

$$P[d_p(\phi(x), \phi(x')) \leq 2\sqrt[p]{d}L * d_p(x,x')] \geq 1 - \alpha \tag{10}$$

Since $P[d_p(\phi(x), \phi(x')) \leq 2\sqrt[p]{d}L * d_p(x,x')]$ in equation 10 defines $A_{\lambda,r}$ for $\lambda \geq 2\sqrt[p]{d}L$, this concludes the proof.

$\square$

---

[1] $d_p(x,y) = \sqrt[p]{\sum_i^d ||x_i - y_i||^p} \leq \sqrt[p]{\sum_i^d \max_i ||x_i - y_i||^p} = \sqrt[p]{d} \max_i d_p(x_i, y_i)$

**Corollary 1.1.** *If the prediction function $f$ is locally $L-$lipschitz at radius $r$ then shapley explainers are $\lambda-$astute for radius $r \geq 0$ for $\lambda \geq 2 \sqrt[p]{d}L$*

*Proof.* Note that definition 3 reduces to the definition of deterministic lipschitz if $\alpha = 0$. Which means equation 10 will be true with probability 1. Which concludes the proof. □

### 3.1.2 ASTUTENESS OF RISE

RISE determines feature explanation for the $i^{th}$ feature by sampling subsets of features and then calculating the mean value of the prediction function when feature $i$ is included in the subset. RISE feature attribution for a given point $x$ and feature $i$ for a prediction function $f$ can be written as:

$$\phi_i(x) = \mathbb{E}_{p(z|z_i=1)}[f(x \odot z)] \tag{11}$$

The following theorem establishes the bound on $\lambda$ for *explainer astuteness* of RISE in relation to the Lipschitzness of blackbox prediction function.

**Theorem 2.** *(Astuteness of RISE) For a given $r \geq 0$ and $0 \leq \alpha \leq 1$, and a trained predictive function $f$ that is locally probabilistic lipschitz with a constant $L$ with radius $r$ and probability at least $1 - \alpha$. Then for RISE explainer we have the astuteness $A_{r,\lambda} \geq 1 - \alpha$, for $\lambda \geq \sqrt[p]{d}L$.*

*Proof.* (Sketch, full proof in Appendix A)

Given input $x$ and another input $x'$ s.t. $d(x, x') \leq r$, using equation 11 we can write

$$
\begin{aligned}
d_p(\phi_i(x), \phi_i(x')) &= d_p(\mathbb{E}_{p(z|z_i=1)}[f(x \odot z)], \mathbb{E}_{p(z|z_i=1)}[f(x' \odot z)]) \\
&= ||\mathbb{E}_{p(z|z_i=1)}[f(x \odot z)] - \mathbb{E}_{p(z|z_i=1)}[f(x' \odot z)]||_p \\
&= ||\mathbb{E}_{p(z|z_i=1)}[f(x \odot z) - f(x' \odot z)]||_p
\end{aligned} \tag{12}
$$

Using Jensen's inequality on R.H.S followed by the fact that $E[f] \leq \max f$

$$d_p(\phi_i(x), \phi_i(x')) \leq \max_z d_p(f(x \odot z), f(x' \odot z)) \tag{13}$$

Using the fact that $f$ is probabilistic Lipschitz gives us and using $d_p(\phi(x), \phi(x') \leq \sqrt[p]{d} * \max_i d_p(\phi_i(x), \phi_i(x'))$ gives us,

$$P[d_p(\phi(x), \phi(x') \leq \sqrt[p]{d}L * d_p(x, x')] \geq 1 - \alpha \tag{14}$$

Since $P[d_p(\phi(x), \phi(x') \leq \sqrt[p]{d}L * d_p(x, x')]$ defines $A_{\lambda,r}$ for $\lambda \geq \sqrt[p]{d}L$, this concludes the proof. □

**Corollary 2.1.** *If the prediction function $f$ is locally $L-$Lipschitz at radius $r \geq 0$, then RISE explanations are $\lambda-$astute for radius $r$ and $\lambda \geq \sqrt[p]{d}L$*

*Proof.* Same as proof for Corollary 1.1. □

### 3.1.3 ASTUTENESS OF "REMOVE INDIVIDUAL" EXPLAINERS

Within the framework of feature removal explainers, a sub-category is the explainers that work by removing a single feature from the set of all features and calculating feature attributions based on change in prediction that result from removing that feature. This category includes Occlusion, CXPlain (Schwab & Karlen, 2019), PredDiff (Zintgraf et al., 2017) Permutation tests (Strobl et al., 2008), and feature ablation explainers (Lei et al., 2018).

Remove individual explainers determine feature explanations for the $i^{th}$ feature by calculating the difference in prediction with and without that feature included for a given point $x$. Let $z_{-i} \in 0, 1^d$ represent a binary vector with $z_i = 0$, then the explanation for feature $i$ can be written as:

$$\phi(x_i) = f(x) - f(x \odot z_{-i}) \tag{15}$$

**Theorem 3.** *(Astuteness of Remove individual explainers) Consider a given $r \geq 0$ and $0 \leq \alpha \leq 1$ and a trained predictive function $f$ that is locally probabilistic Lipschitz with a constant $L$ with radius $r$ measured using $d_p(.,.)$ induced by the $p$-norm and probability at least $1 - \alpha$. Then for Remove individual explainers, we have the astuteness $A_{r,\lambda} \geq 1 - \alpha$, for $\lambda \geq 2 \sqrt[p]{d}L$, where $d$ is the dimensionality of the data.*

*Proof.* (Sketch, full proof in Appendix A) By considering another point $x'$ such that $d_p(x, x') \leq r$ and equation 15 we get,

$$d_p(\phi(x_i), \phi(x'_i)) = d_p(f(x) - f(x \odot z_{-i}), f(x') - f(x' \odot z_{-i})) \tag{16}$$

then following the exact same steps as the proof for Theorem 1 i.e. writing the right hand side in terms of $p$-norm, utilizing triangular inequality, and the definition of probabilistic Lipschitzness leads us to the desired result. □

**Corollary 3.1.** *If the prediction function $f$ is locally $L-$Lipschitz at radius $r \geq 0$, then remove individual explanations are $\lambda-$astute for radius $r$ and $\lambda \geq 2 \sqrt[p]{d}L$.*

*Proof.* Same as proof for Corollary 2.1. □

## 3.2 IMPLICATIONS

The above theoretical results all provide the same critical implication, that is, explainer astutness is lower bounded by the Lipschitzness of the prediction function. This means that black-box classifiers that are locally smooth (have a small $L$ at a given radius $r$) lend themselves to probabilistically more robust explanations. While there has already been work enforcing Lipschitzness on neural networks through regularization (Gouk et al., 2021), it has primarily been with the goal of improving performance. Our results provide motivation for the same except by showing that doing so will result in a higher lower bound on the astuteness of explanations on such Lipschitz controlled classifiers.

## 4 EXPERIMENTS

To demonstrate the validity of our experimental results, we perform a series of experiments on 7 datasets. We train four different classifiers on each of these datasets, and then explain the decisions of these classifiers using three explainer methods. The details are as follows:

**Datasets**. We use five simulated datasets based on datasets introduced by Chen et al. (2018). The first three, labelled *XOR*, *Orange Skin*(OS), and *Nonlinear Additive*(NA) are generated such that the ground truth predictions depend on the same features for all points. The last two labelled *Switch* and *Switch++* are constructed such that the ground truth predictions depend on different features in different regions of the input space. More details about these datasets are provided in Appendix B.

In addition to the above simulated datasets, we also provide evaluations on two real datasets from the UCI Machine learning repository (Asuncion & Newman, 2007) for binary classification. *Rice* (Cinar & Koklu, 2019) consists of 3810 samples of rice grains of two different varieties (*Cammeo* and *Osmancik*). 7 morphological features are provided for each sample. *Telescope* (Ferenc et al., 2005) consists of 19000+ Monte-Carlo generated samples to simulate registration of high energy gamma particles in a ground-based atmospheric Cherenkov gamma telescope using the imaging technique. Each sample is labelled as either background or gamma signal and consists of 10 features.

**Classifiers**. For each dataset we train the following four classifiers; **2layer**: A two-layer MLP with ReLU activations. For simulated datasets each layer has 200 neurons, while for the 2 real datasets we use 32 neurons in each layer. **4layer**: A four-layer MLP with ReLU activations, with the same number of neurons per layer as *2layer*. **linear**: A linear classifier constructed by removing the non-linear activations from the *2layer* classifier. **svm**: A support vector machine with Gaussian kernel.

The idea here is that each of these classifiers will have different probabilistically Lipschitz behavior, and that can be used to lower bound the explainer astuteness when explaining each of these classifiers according to our theoretical results.

**Explainers**. We evaluate 3 explainers here that are representative of the 3 theorems provided in **SHAP** (Lundberg & Lee, 2017) serves as Representative of Theorem 1. We use the gradient based

approximation for the neural-network classifiers and the kernel shap approximation for SVM. Both are included in the implementation provided by the authors[2]. **RISE** (Petsiuk et al., 2018)serves as representative method for Theorem 2. The official implementation provided by the authors is primarily for image datasets [3]. We adapt this for tabular datasets. **CXPlain** (Schwab & Karlen, 2019) serves as representative method for Theorem 3. We use the official implementation provided by the authors [4].Section 3:

### 4.1 ESTIMATING PROBABILISTIC LIPSCHITZNESS AND LOWER BOUND FOR ASTUTENESS

To demonstrate the connection between explainer astuteness and probabilistic lipchitzness as alluded to by our theory we need to estimate probabilistic Lipschitzness for classifiers. In our experiments we achieve this by by empirically estimating the $\mathbb{P}_{x,x'} \sim D$ (equation 3) for a range of values of $L \in \{0, 1\}$ incremented at $0.1$. We do this for each classifier and for each dataset $D$ and set $r$ as median of pairwise distance for all training points. According to equation 3 this gives us an upperbound on $1 - \alpha$ i.e. we can say that for a given $L, r$ the classifier is Lipschitz with probability at least $1 - \alpha$.

We can use the estimates for probabilistic Lipschitzness to predict astuteness using our theorems. We do this by noting that our theorems imply that for $\lambda \geq CL\sqrt{d}$ explainer astuteness is at least $1 - \alpha$. This means we can simply multiply the range of Lipschitz constant $L$ with $C\sqrt{d}$ and for $\lambda$ greater or equal to that value we can guarantee that explainer astuteness should be lower bounded by $1 - \alpha$. This is how we arrive at the dashed lines in Figure 2.

### 4.2 RESULTS

The goal of our experiments is to empirically show that we can use the probabilistic Lipschitzness of classifiers to predict the lower bound for explainer astuteness. If we can plot astuteness lower bound predicted by our theory against observed astuteness for different values of epsilon and the observed astuteness curves are indeed shown to be lower bounded by the predicted astuteness curves, which are the desired result.

To this end, for each of the seven datasets we first train the four classifiers. We then measure the probabilistic Lipschitzness of each of these classifiers for each dataset. Afterwards we explain the predictions of these classifiers using each of the three explainer methods listed above. This results in 3 subplots for each dataset, one for each explainer, as shown in Figure 2, for *Nonlinear Additive*, *Switch++*, *Rice* and *Telescope* datasets. Plots for all datasets can be seen in Figure 3.

Each subplot in Figure 2 presents two types of curves, the dashed curves represent the predicted lower bound on explainer astuteness given a classifier, as described in Section 4.1. The solid curves are the actual estimations of explainer astuteness using Definition 2. According to our theoretical results, at a given $\lambda$ the estimated explainer astuteness (solid curves) should stay above the predicted astuteness (dashed curves) based on the Lipschitzness of classifiers. As Figure 2 demonstrates, this is indeed the case. Notice that some deviations from this can be expected since the theoretical results assume the ideal scenario for each of the methods and assume that the probabilities can be computed exactly. In practice e.g. while our theorems use the theoretical definition of SHAP, the implementation uses a Gradient Explainer approximation provided by Lundberg & Lee (2017). Similarly, all probabilities are calculated using empirical estimates using the available training points. Table 1 presents the results in tabular form for all datasets. It shows the difference between the AUC under the estimated astuteness curves ($\mathbf{AUC}$) and the AUC under the predicted lower bound ($\mathbf{AUC_{lb}}$). This number captures the average tightness of the lowerbound over a range of $\lambda$ values. This number should be non-negative, which we observe it is, except in one case where it's a small negative value due to estimation errors. We also observe that given at least some of the values in this table are not close to $0$ our proposed lower bound can be improved upon in the future to make it tighter.

### 4.3 CONCLUSION AND FUTURE WORK

In this paper we formally defined *explainer astuteness* which captures the probability that a given explainer will assign similar explanations to similar points. We theoretically prove that this explainer astuteness is proportional to the *probabilistic Lipschitzness* of the black-box function that is being

---

[2]https://github.com/slundberg/shap
[3]https://github.com/eclique/RISE
[4]https://github.com/d909b/cxplain

explained. As probabilistic Lipschitzness captures local smoothness properties of a function, this result suggests that enforcing smoothness on black-box models can lend these models to more robust explanations. By way of future work we observe that our experimental results suggest that our predicted lower bound can be tightened further. We intend to pursue this in future research.

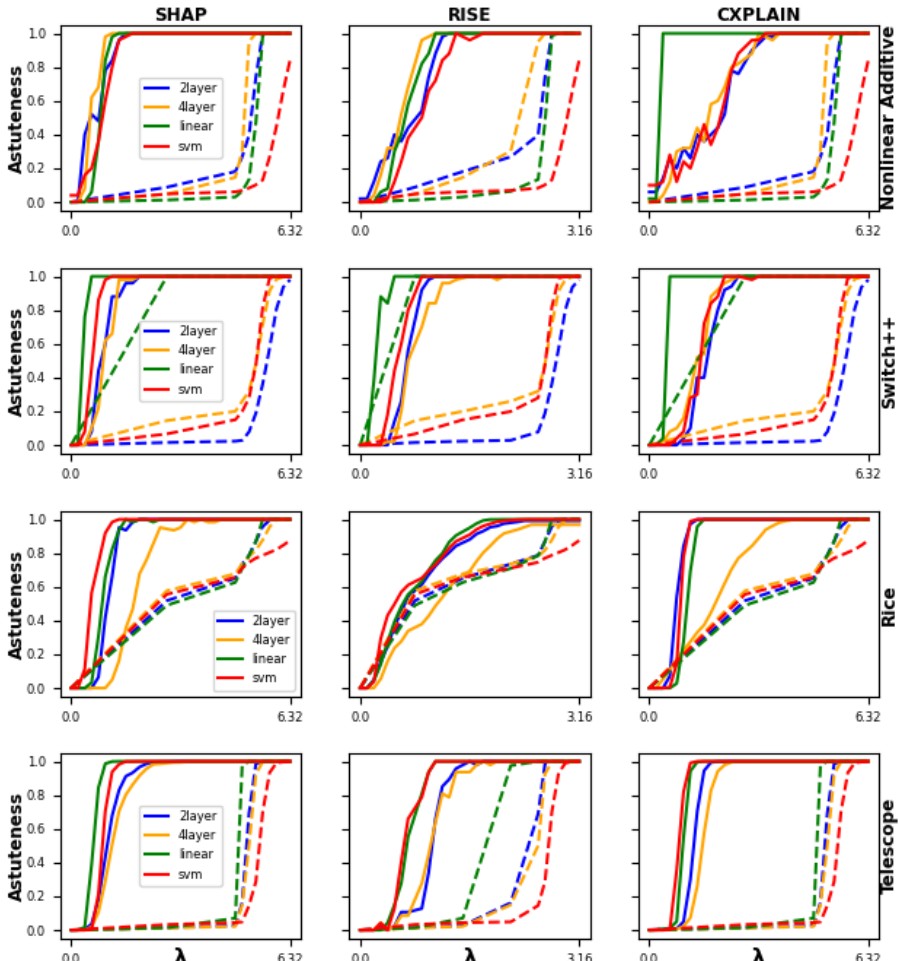

Figure 2: This figure experimentally shows the implication of our theoretical results. Given the datasets and the four classifiers trained on each of these datasets, we observe that the explainer astuteness for SHAP, RISE and CXPLAIN is lower bounded by the astuteness predicted by our theoretical results given a value of $\lambda$ along the x-axis. The predicted lower bound is depicted by dashed lines, while solid lines depict the actual estimate of explainer astuteness. See Figure 3 for plots for all seven datasets.

Table 1: $\mathbf{AUC} - \mathbf{AUC_{lb}}(\downarrow)$. The observed AUC is lower bounded by the predicted AUC. The difference between the two is always $\geq 0$. This represents the average tightness of the lower bound.

| | 2layer | | | 4layer | | | linear | | | svm | | |
|---|---|---|---|---|---|---|---|---|---|---|---|---|
| Datasets | SHAP | RISE | CXPlain | SHAP | RISE | CXPlain | SHAP | RISE | CXPlain | SHAP | RISE | CXPlain |
| XOR | .049 | .050 | -.013 [5] | .049 | .050 | .003 | .049 | .050 | .031 | .642 | .578 | .589 |
| OS | .585 | .477 | .551 | .489 | .415 | .426 | .043 | .017 | .043 | .761 | .628 | .732 |
| NA | .359 | .289 | .318 | .285 | .216 | .244 | .452 | .391 | .474 | .742 | .653 | .708 |
| Switch | .053 | .053 | .003 | .086 | .083 | .039 | .043 | .028 | .034 | .557 | .472 | .524 |
| Switch++ | .618 | .557 | .590 | .415 | .342 | .398 | .041 | .025 | .035 | .433 | .377 | .399 |
| Rice | .159 | .084 | .171 | .138 | .031 | .138 | .168 | .102 | .162 | .265 | .192 | .254 |
| Telescope | .324 | .213 | .317 | .345 | .244 | .333 | .223 | .149 | .211 | .501 | .439 | .504 |

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

## A DETAILED PROOFS

We include the detailed proofs for Theorems 2 and 3 here.

*Proof.* (For Theorem 2)

Given input $x$ and another input $x'$ s.t. $d(x, x') \leq r$, using equation 11 we can write

$$
\begin{aligned}
d_p(\phi_i(x), \phi_i(x')) &= d_p(\mathbb{E}_{p(z|z_i=1)}[f(x \odot z)], \mathbb{E}_p(z|z_i = 1)[f(x' \odot z)]) \\
&= ||\mathbb{E}_{p(z|z_i=1)}[f(x \odot z)] - \mathbb{E}_p(z|z_i = 1)[f(x' \odot z)]||_p \\
&= ||\mathbb{E}_{p(z|z_i=1)}[f(x \odot z) - f(x' \odot z)]||_p
\end{aligned}
\tag{17}
$$

Using Jensen's inequality on R.H.S,

$$
d_p(\phi_i(x), \phi_i(x')) \leq \mathbb{E}_{p(z|z_i=1)}[||f(x \odot z) - f(x' \odot z)||_p]
\tag{18}
$$

Using the fact that $E[f] \leq \max f$,

$$
\begin{aligned}
d_p(\phi_i(x), \phi_i(x')) &\leq \max_z ||f(x \odot z) - f(x' \odot z)||_p \\
&= \max_z d_p(f(x \odot z), f(x' \odot z))
\end{aligned}
\tag{19}
$$

Using the fact that $f$ is probabilistic Lipschitz with some constant $L \geq 0$, and $d_p(x \odot z, x' \odot z) \leq d_p(x, x'), \forall z$. Then using the definition of probabilistic Lipschitz we get,

$$
P(\max_z d_p(f(x \odot z), f(x' \odot z)) \leq L * d(x, x') \geq 1 - \alpha
\tag{20}
$$

Using this in equation 19 gives us,

$$
P[d_p(\phi_i(x), \phi_i(x')) \leq L * d(x, x')] \geq 1 - \alpha
\tag{21}
$$

Note that equation 21 is true for each feature $i \in \{1, ..., d\}$. To conclude the proof note that $d_p(\phi(x), \phi(x') \leq \sqrt[p]{d} * \max_i d_p(\phi_i(x), \phi_i(x'))$. Utilizing this with equation 21 leads us to

$$
P[d_p(\phi(x), \phi(x') \leq \sqrt[p]{d}L * d_p(x, x')] \geq 1 - \alpha
\tag{22}
$$

Since $P[d_p(\phi(x), \phi(x') \leq \sqrt[p]{d}L * d_p(x, x')]$ defines $A_{\lambda,r}$ for $\lambda \geq \sqrt[p]{d}L$, this concludes the proof. $\square$

*Proof.* (For Theorem 3)

By considering another point $x'$ such that $d_p(x, x') \leq r$ and equation 15 we get,

$$
d_p(\phi(x_i), \phi(x_i')) = d_p(f(x) - f(x \odot z_{-i}), f(x') - f(x' \odot z_{-i}))
\tag{23}
$$

using the fact that $d_p(x, y) = ||x - y||_p$ where $||.||_p$ is the $p$-norm, the RHS gives us,

$$
d_p(\phi_i(x), \phi_i(x')) = ||f(x) - f(x \odot z_{-i}) - f(x') + f(x' \odot z_{-i})||_p
\tag{24}
$$

using triangular inequality,

$$
d_p(\phi_i(x), \phi_i(x')) \leq ||f(x) - f(x')||_p + ||f(x' \odot z_{-i}) - f(x \odot z_{-i})||_p
\tag{25}
$$

w.l.o.g assuming the first term on the right is bigger than the second term

$$d_p(\phi_i(x), \phi_i(x')) \leq 2||f(x) - f(x')||_p = 2d_p(f(x), f(x')) \tag{26}$$

using the fact that $f$ is probabilistic Lipschitz get us,

$$P[d_p(\phi_i(x), \phi_i(x')) \leq 2Ld_p(x, x')] \geq 1 - \alpha \tag{27}$$

to conclude the proof note that $d_p(\phi(x), \phi(x')) \leq \sqrt[p]{d} * \max_i d_p(\phi_i(x), \phi_i(x'))$, which gives us,

$$P[d_p(\phi(x), \phi(x')) \leq 2\sqrt[p]{d}L * d_p(x, x')] \geq 1 - \alpha \tag{28}$$

$\square$

## B    DATASET DETAILS

- **XOR**: A dataset with 2 classes. The input data is generated from a 10-dimensional standard Gaussian distribution. The class probabilities are generated proportional to $\exp\{X_1 X_2\}$. That is only the first two features are important in generating predictions for *all* data points.

- **Orange-skin**: The input data is again generated from a 10-dimensional standard Gaussian distribution. The ground truth class probabilities are proportional to $\exp\{\sum_{i=1}^4 X_i^2 - 4\}$. In this case the first 4 features are important globally for *all* data points.

- **Nonlinear-additive**: Similar to *Orange-skin* dataset except the ground trugh class probabilities are proportional to $\exp\{-100 \sin 2X_1 + 2|X_2| + X_3 + \exp\{-X_4\}\}$, and therefore each of the 4 important features for prediction are nonlinearly related to the prediction itself.

- **Switch**: This simulated dataset is specifically for instancewise feature explanations. For the input data feature $X_1$ is generated by a mixture of Gaussian distributions centered at $\pm 3$. If $X_1$ is generated from the Gaussian distribution centered at $+3$, $X_2$ to $X_5$ are used to generate the prediction probabilities according to the *Orange skin* model. Otherwise $X_6$ to $X_9$ are used to generate the prediction probabilities according to the *Nonlinear-additive* model.

- **Switch++**: This is a modified version of the *Switch* dataset where instead of two classes there are 4 classes with varying weights.

## C    ADDITIONAL RESULTS

Table 2 shows the normalized AUC for the estimated explainer astuteness and the predicted AUC based on the predicted lower bound curve. As expected the predicted AUC lower bounds the estimated AUC.

Figure 3 shows the same plots as shown in Figure 2 but includes all datasets.

Table 2: **Observed AUC and (Predicted AUC)**. The observed AUC is lower bounded by the predicted AUC and so the observed AUC should always be higher than the predicted AUC. The AUC values are normalized between 0 and 1.

| Datasets | 2layer | | | | 4layer | | | | linear | | | | svm | | | |
|---|---|---|---|---|---|---|---|---|---|---|---|---|---|---|---|---|
| | **SHAP** | **RISE** | **CXP** | **(LB)** | **SHAP** | **RISE** | **CXP** | **(LB)** | **SHAP** | **RISE** | **CXP** | **(LB)** | **SHAP** | **RISE** | **CXP** | **(LB)** |
| XOR | 1.00 | 1.00 | .937 | (.950) | 1.00 | 1.00 | .954 | (.950) | 1.00 | .999 | .982 | (.950) | .975 | .912 | .922 | (.333) |
| OS | .954 | .847 | .920 | (.369) | .969 | .896 | .906 | (.480) | .994 | .967 | .994 | (.950) | .945 | .813 | .917 | (.184) |
| NA | .978 | .909 | .936 | (.618) | .981 | .926 | .940 | (.696) | .972 | .912 | .994 | (.520) | .971 | .883 | .937 | (.229) |
| Switch | .998 | .996 | .948 | (.945) | .996 | .988 | .948 | (.909) | .994 | .978 | .988 | (.950) | .969 | .885 | .936 | (.412) |
| Switch++ | .969 | .908 | .941 | (.350) | .967 | .894 | .950 | (.552) | .992 | .974 | .986 | (.950) | .982 | .926 | .947 | (.548) |
| Rice | .962 | .886 | .974 | (.803) | .932 | .824 | .932 | (.793) | .968 | .901 | .962 | (.800) | .981 | .906 | .970 | (.715) |
| Telescope | .962 | .863 | .954 | (.637) | .955 | .863 | .944 | (.610) | .980 | .906 | .967 | (.756) | .969 | .909 | .972 | (.467) |

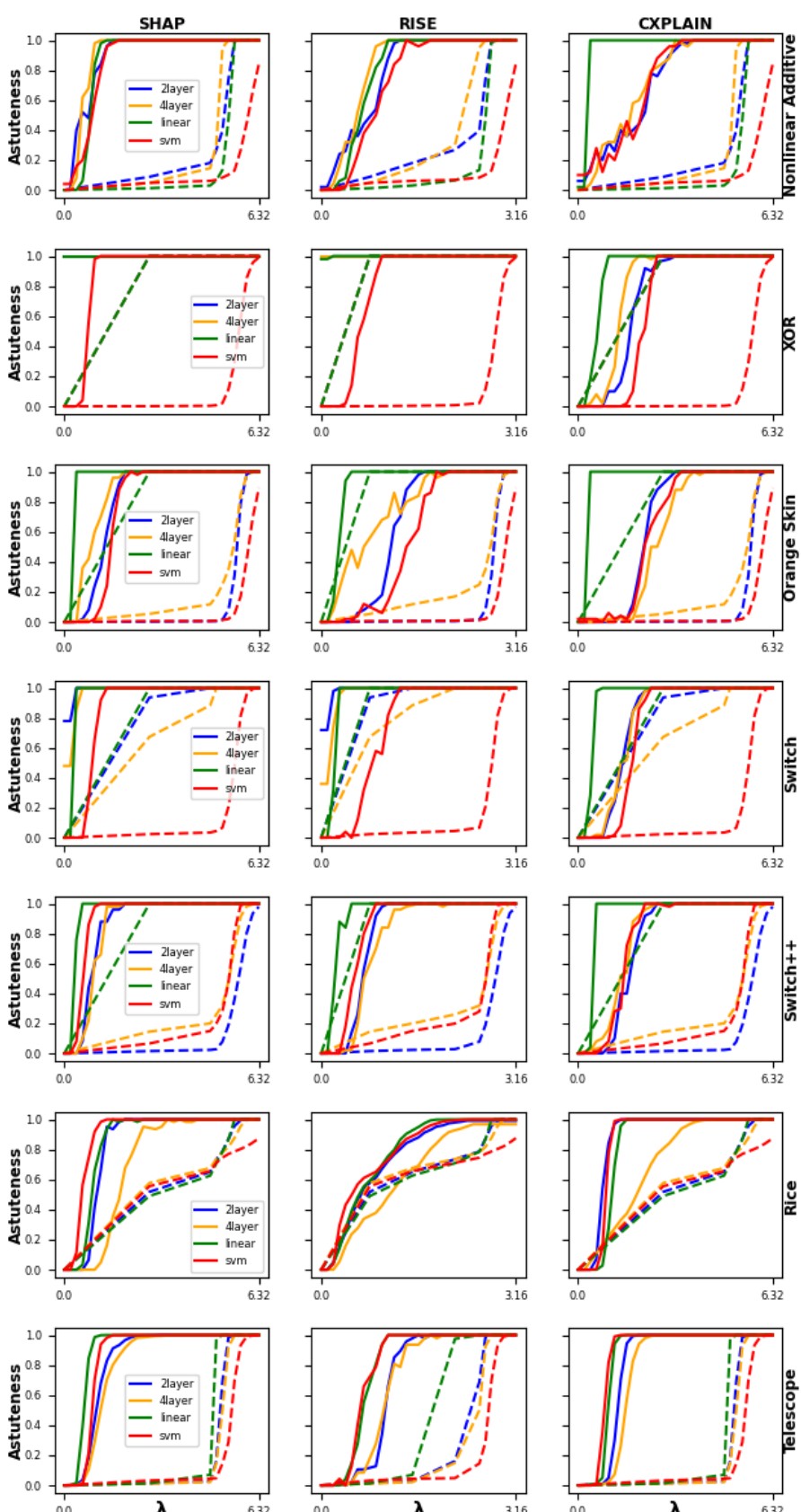

Figure 3: This figure experimentally shows the implication of our theoretical results. Given the seven datasets and the four classifiers trained on these each of these datasets, we observe that the explainer astuteness for SHAP, RISE and CXPLAIN is lower bounded by the astuteness predicted by our theoretical results given a value of $\lambda$. The predicted lower bound is depicted by dashed lines, while solid lines depict the actual estimate of explainer astuteness.

