# OpenReview forum: "Analyzing the Effects of Classifier Lipschitzness on Explainers"
_ICLR.cc/2022/Conference — ICLR 2022 Submitted_

### Official Review · Reviewer_ftCD · 2021-10-27

**Correctness:** 3
**Technical Novelty And Significance:** 2
**Empirical Novelty And Significance:** 2
**Recommendation:** 5
**Confidence:** 4

**Main Review:**

The paper essentially talks about the robustness of an explanation method that has been extensively studied in the XAI literature [1,2. 3]. Given below is a list of the papers' strengths and weaknesses:

Strengths:
1. Given the Lipschitzness of the prediction function, the paper provides lower bound guarantees on the astuteness of different explanation methods.
2. It analyzes three types of explanation methods: i) Shapley value-based, ii) explainers that simulate the mean effect of features, and iii) explainers that simulate individual feature removal.
3. Empirical evaluations using real and synthetic datasets demonstrate that the lower bound holds in practice.

Weaknesses:
1. Alvarez-Mellis et al. [1] defined that an explanation satisfies robustness if similar inputs do not result in substantially different explanations. The explainer astuteness (defined by the authors) is similar to the above definition of robustness. However, both robustness and astuteness do not consider the model behavior while defining similarity. The underlying assumption is that the model prediction remains the same for similar inputs, but we should also consider that the model can output the same predictions using different logic paths. It would be great if the authors can comment on this, or will the astuteness formulation change on considering this.
2. The paper is hastily written with errors in proofs: i) In the proof of Theorem 1, the last two $f(\cdot)$ terms of the first two equations should be $f(x' \odot  z_{-i})$. ii) In Equation 5, the sign associated with $f(x' \odot  z_{-i})$ should be $+$, and iii) typos in Equation 12.
3. The assumption made in Equation 6 for taking the maximum is unclear. It would be helpful if the authors can expand that.
4. In the experiments, the authors mention that "some deviations can be expected since the theoretical results assume the ideal scenario for each of the methods and assume that the probabilities can be computed exactly, which is not the case in practice" -- what do the authors mean by the ideal scenario? and if the calculated probabilities are not exact, then how reliable are the bounds?

[1]  Alvarez-Melis, David, and Tommi S. Jaakkola. "On the robustness of interpretability methods." ICML WHI. 2018.

[2] David Alvarez-Melis, Tommi S. Jaakkola, "Towards Robust Interpretability with Self-Explaining Neural Networks". In NeurIPS, 2018.

[3] Lakkaraju, Himabindu, Nino Arsov, and Osbert Bastani. "Robust and stable black box explanations." In ICML, 2018.

**Summary Of The Paper:**

The paper focuses on the reliability of explainers, where an explainer should give similar explanations for similar data inputs. To this end, the authors introduce and define "explainer astuteness", which leverages the concept of "probabilistic Lipschitzness" (the probability of local smoothness of a predictor/classifier function). The paper explores the reliability of different types of explainers by analyzing the connection between the robustness of explanation methods and the smoothness of the black-box predictor function they are trying to explain. Finally, the authors provide a theoretical way to connect the explainer astuteness to the probabilistic Lipschitzness of the black-box function and explore how the smoothness of the black-box function impacts an explainers' astuteness.

**Summary Of The Review:**

The paper presents theoretical guarantees for different classes of explanation methods. However, the notion of stability/robustness/astuteness has been extensively studied in the existing literature and the novelty of the work is thus a bit unclear.

---

> ### Author Response · Authors · 2021-11-18
> **Response to Reviewer ftCD**
>
> We appreciate you taking the time to go through the paper and for the very helpful critique. Please find below responses to each of the points you raised. Please feel free to ask any further questions you might have.
>
> *Both robustness and astuteness do not consider the model behavior while defining similarity. The underlying assumption is that the model prediction remains the same for similar inputs, but we should also consider that the model can output the same predictions using different logic paths. It would be great if the authors can comment on this, or will the astuteness formulation change on considering this.*
>
> **We agree that ideally explainers should take into account logical paths in addition to just the predictions. However the explainers currently in use and investigated here do not do that. As such, this is a limitation of current explainers rather than this method. The astuteness formulation is general and will stand in that case as well, except \phi(x) would then be a function of the logical path in addition to the function output.**
>
> *The paper is hastily written with errors in proofs: i) In the proof of Theorem 1, the last two f(⋅)  terms of the first two equations should be f(x′⊙z−i). ii) In Equation 5, the sign associated with f(x′⊙z−i)  should be + and iii) typos in Equation 12.*
>
> **changes have been made and the typos have been fixed**
>
> *The assumption made in Equation 6 for taking the maximum is unclear. It would be helpful if the authors can expand that.*
>
>  **Added “We can replace each value inside the sums in equation 6 with the maximum value across either sums. Doing so would still preserve the inequality in equation 6, as the sum of $n$ values is always less than the maximum among those summed $n$ times.” in the revised version.**
>
> *"some deviations can be expected since the theoretical results assume the ideal scenario for each of the methods and assume that the probabilities can be computed exactly, which is not the case in practice" -- what do the authors mean by the ideal scenario? and if the calculated probabilities are not exact, then how reliable are the bounds?*
>
> **We have now clarified this in the paper. We were referring to the fact that the theorems use the ideal formulation for each method, e.g. theorems covering SHAP use the exact formulation for shapley values, but the implementations available use the Gradient Explainer approximation. In addition, the probabilities in Eq(2) and Eq(3) are estimated using samples, the goodness of this empirical estimate depends on how many training points are available to approximate the probability.**
>
> *The notion of stability/robustness/astuteness has been extensively studied in the existing literature and the novelty of the work is thus a bit unclear.*
>
> **We agree and acknowledge that these notions have been extensively studied with respect to classifiers and prediction functions. Bhattacharjee et al. (2020)’s work on classifier astuteness in fact served as a motivation for us to explore explainer astuteness.**
>
> **There are only a few works on looking at theoretical properties of explainers. To the best of our knowledge this is the first work that provides a clear formulation for explainer astuteness and theoretically demonstrates its connection to classifier smoothness.**

---

### Official Review · Reviewer_Np65 · 2021-10-30

**Correctness:** 2
**Technical Novelty And Significance:** 2
**Empirical Novelty And Significance:** 2
**Recommendation:** 3
**Confidence:** 5

**Main Review:**

Strengths:
+ very important problem to study. The robustness of explanation problems has been studied by prior work, but there is a lack of theoretical analysis from the perspective of function smoothness.
+ the proofs are detailed and correct.
+ covers some of the representative explanation methods, such as SHAP.
+ data and codes are released.

Weaknesses:
- The proof does not bring much new insight that are relevant to explanation robustness. There two sub-problems that are important but not addressed: 1) what's the worst-case change in the explanations (that would require a upper-bound over \epsilon, rather than a lower-bound); 2) the individual explanation element are treated independently but in practice, the features are ranked and only a small number of them are selected as an explanations, and a small change in explanation vector may lead to rather different top-k salient features.

- experimental sections are not comprehensive and vague. For example, it is not clear how to estimate the Lipschitz constant of a neural network. It is also not clear why the predicted astuteness should lower-bound the estimated one. In fact, the definition of the predicted/estimated astuteness is not given. How the experiment in Figure 2 support the theory is not explained clearly. Lastly, from Figure 2, the bounds seem quite loose: will they be useful in practice, and how they can be used?

**Summary Of The Paper:**

This paper studies the problem of robustness of local explanations of a classifier at some input x. Explanations can be generated by several methods: Shapley-value-based, mean-effect-based, and removal-based. Since the explanation methods use the output of the classifier f, the change in the explanations can be related to the change in the output of f, and the latter is quantified by the Lipschitz constant of f.
Lower bounds of \epsilon that bounds the change in the explanation in "explanation astuteness" is proved for three example explanation methods. This gives the "best-case" robustness in the explanations given perturbation in the input x. Experiments on 7 datasets evaluated the tightness of a low-bound of "explanation astuteness", which is the probability that two close inputs x and x' have their explanations \phi(x) and \phi(x') differ by no more than some upper-bound related to d(x,x').

**Summary Of The Review:**

Problem studied are good, but theory and experiments have room for improvement.

---

> ### Author Response · Authors · 2021-11-18
> **Response to Reviewer NP65**
>
> Thank you for appreciating the strengths of the paper and the constructive criticism. We believe responding to these comments and incorporating suggested changes will improve the quality of this work. Please find pointwise responses to the issues you raised below. Please feel free to ask any further questions you might have.
>
> *There two sub-problems that are important but not addressed: 1) what's the worst-case change in the explanations (that would require a upper-bound over \epsilon, rather than a lower-bound); 2) the individual explanation element are treated independently but in practice, the features are ranked and only a small number of them are selected as an explanations, and a small change in explanation vector may lead to rather different top-k salient features.*
>
> **1. We agree that finding both an upper bound and a lower bound would be very useful. However, we argue that the lower bound on epsilon is in itself useful as well. We think that some of the message is being lost due to our choice of notation (as \epsilon is traditionally used for “error”), hence we are changing \epsilon to \lambda in the paper. The lower bound on \lambda tells us that, given a predictor with probabilistic Lipschitz constant L with probability at least 1 - \alpha, we can expect explainers to have astuteness at least 1- \alpha for \lambda \geq CL\sqrt{d}. This translates to a guarantee that the explainer astuteness will be < 1 - \alpha when considering smaller \lambda values. Ideally we would want explainers to be highly astute for small values of \lambda; our theorems demonstrate that desirable property is limited by the Lipschitzness of the predictor function itself.
> 2. Our definitions and theorems will still hold even if only a subset of the top-k ranked features are considered. The non top-k features can simply be zeroed out with a mask.**
>
> *It is not clear how to estimate the Lipschitz constant of a neural network. It is also not clear why the predicted astuteness should lower-bound the estimated one. In fact, the definition of the predicted/estimated astuteness is not given. How the experiment in Figure 2 support the theory is not explained clearly. Lastly, from Figure 2, the bounds seem quite loose: will they be useful in practice, and how can they be used?*
>
> **Estimating the deterministic Lipschitz constant of a neural network is indeed not straightforward. However our work uses probabilistic Lipchitzness; we describe in subsection 4.1 (added in revised version) in the experiments how we estimate this. To summarize, we empirically estimate the probability in equation (3) for a range of L, for fixed ‘r’ as the median pairwise distance of all available data points. As L increases, this probability goes to 1. For any given value of L you can say that the classifier is Lipschitz with constant L and probability as calculated in eq(3) for the neighborhood of radius ‘r’.
> As we acknowledge in the conclusions section, we agree that the bounds in Figure 2 are quite loose, however they still demonstrate the main message of the paper and the theorems is sound by showing that predicted astuteness (which we now also explain clearly in subsection 4.1) indeed lower bounds the observed astuteness. We intend to work on tightening these bounds as part of future work.The practical utility and message of our theoretical work is that we now know that explainer astuteness is related to predictor Lipschitzness.  Our theoretical result provides a motivation for ML researchers to improve predictor Lipschitzness to ensure better explainer astuteness (robustness). We have added a clear emphasis on this point in the Introduction section of the revised version.**

---

### Official Review · Reviewer_ARPU · 2021-11-01

**Correctness:** 4
**Technical Novelty And Significance:** 2
**Empirical Novelty And Significance:** 2
**Recommendation:** 5
**Confidence:** 3

**Main Review:**

The definition of astuteness introduced as a criterion for assessing the reliability of explanations seems to be a natural and useful definition. It is also interesting that lower bounds of astuteness can be derived for well-known explanatory methods. On the other hand, the derivation is elemental and does not lead to surprising results or useful results that can be used to improve the reliability of explanations. The result that the robustness of the explainer is bounded by the Lipschitz constant of the prediction function also seems obvious, since useful explanatory methods are naturally required to be robust against perturbation.

**Summary Of The Paper:**

In this study, a metric called Astuteness is introduced as a criterion to evaluate the reliability of explanations. Intuitively, if the distance between explanations given for two samples that are within a certain distance is also within a certain distance, then the explanation has astuteness. This definition can be interpreted as defining the robustness of an explanation. More specifically, it relates the distance in the sample space to the distance in the explanation space by probabilistic Lipschitzness. In this study, the authors show that for some certain classes of explanatory methods, such as SHAP and CXplain, the probabilistic lower bound of the astuteness of an explanation depends on the square root of the number of feature dimensions D and the Lipschitz constant L. The reliability of this probabilistic lower bound is also evaluated experimentally with several datasets.

**Summary Of The Review:**

Although the proposed definition is natural and useful as a criterion for evaluating the reliability of explanations, it cannot be said that the analysis using the criterion leads to useful results for improving explanatory methods, and it seems that more detailed analysis and deeper insight into the relationship between explanatory methods and the proposed criterion are needed for publication.

---

> ### Author Response · Authors · 2021-11-18
> **Response to Reviewer ARPU**
>
> Thank you for the constructive review and for appreciating the possible usefulness of our proposed formulation. Please find below responses to each of the points raised in the review. Please feel free to ask any further questions or clarifications you might have.
>
>
> *The result that the robustness of the explainer is bounded by the Lipschitz constant of the prediction function also seems obvious, since useful explanatory methods are naturally required to be robust against perturbation.*
>
> **We agree that useful explanatory methods should be required to be robust against perturbation. Our work is more concerned with proving that enforcing Lipchitzness on classifiers has a direct effect on the robustness of explanations for those classifiers. Our work serves as additional motivation for research focused on enforcing smoothness on neural networks.**
>
>
> *Although the proposed definition is natural and useful as a criterion for evaluating the reliability of explanations, it cannot be said that the analysis using the criterion leads to useful results for improving explanatory methods*
>
> **Our intention in this work was not to improve existing explanatory methods, but instead to prove that improving classifiers by making them more smooth has a direct effect on the explanations provided by these existing methods. Our core message is that smoother classifiers lend themselves to more robust explanations. We have added a clear emphasis on this point in the Introduction section of the revised version.**

---

### Decision · Program_Chairs · 2022-01-20

**Decision:**

Reject

**Comment:**

The paper explores "Astuteness of explainer", to measure reliability of the explanations. There were concerns about the overlap of the proposed work with existing literature.  It was felt that both theory and experiments need more development